# Pneumonia and Related Conditions in Critically Ill Patients—Insights from Basic and Experimental Studies

**DOI:** 10.3390/ijms23179896

**Published:** 2022-08-31

**Authors:** Darya A. Kashatnikova, Maryam B. Khadzhieva, Dmitry S. Kolobkov, Olesya B. Belopolskaya, Tamara V. Smelaya, Alesya S. Gracheva, Ekaterina V. Kalinina, Sergey S. Larin, Artem N. Kuzovlev, Lyubov E. Salnikova

**Affiliations:** 1The Laboratory of Ecological Genetics, Vavilov Institute of General Genetics, Russian Academy of Sciences, Moscow 119991, Russia; 2The Laboratory of Clinical Pathophysiology of Critical Conditions, Federal Research and Clinical Center of Intensive Care Medicine and Rehabilitology, Moscow 107031, Russia; 3The Laboratory of Molecular Immunology, Rogachev National Research Center of Pediatric Hematology, Oncology and Immunology, Moscow 117198, Russia; 4The Resource Center “Bio-Bank Center”, Research Park of St. Petersburg State University, St. Petersburg 199034, Russia

**Keywords:** pneumonia, sepsis, acute respiratory distress syndrome (ARDS), Columbia Open Health Data (COHD), comorbidities, TREC/KREC

## Abstract

Pneumonia is an acute infectious disease with high morbidity and mortality rates. Pneumonia’s development, severity and outcome depend on age, comorbidities and the host immune response. In this study, we combined theoretical and experimental investigations to characterize pneumonia and its comorbidities as well as to assess the host immune response measured by TREC/KREC levels in patients with pneumonia. The theoretical study was carried out using the Columbia Open Health Data (COHD) resource, which provides access to clinical concept prevalence and co-occurrence from electronic health records. The experimental study included TREC/KREC assays in young adults (18–40 years) with community-acquired (CAP) (*n* = 164) or nosocomial (NP) (*n* = 99) pneumonia and healthy controls (*n* = 170). Co-occurring rates between pneumonia, sepsis, acute respiratory distress syndrome (ARDS) and some other related conditions common in intensive care units were the top among 4170, 3382 and 963 comorbidities in pneumonia, sepsis and ARDS, respectively. CAP patients had higher TREC levels, while NP patients had lower TREC/KREC levels compared to controls. Low TREC and KREC levels were predictive for the development of NP, ARDS, sepsis and lethal outcome (*AUC_TREC_* in the range 0.71–0.82, *AUC_KREC_* in the range 0.67–0.74). TREC/KREC analysis can be considered as a potential prognostic test in patients with pneumonia.

## 1. Introduction

Pneumonia is an acute respiratory infection affecting the lungs. In general, pneumonia is classified into community-acquired pneumonia (CAP), which is acquired outside the hospital, and nosocomial pneumonia (NP), which develops 48 h or more after admission. The latter, in turn, includes hospital-acquired pneumonia (HAP) and ventilator-associated pneumonia occurring 48 h or more after endotracheal intubation [1]. The CAP and NP substantially differ in causative microorganisms. The most common microorganisms causing CAP are *Streptococcus pneumoniae* (pneumococcus), respiratory viruses, *Staphylococcus aureus, Haemophilus influenzae, Klebsiella pneumoniae, Mycoplasma pneumoniae* and *Legionella pneumophila.* The most common microorganisms in NP are *Staphylococcus aureus* (including both methicillin-susceptible *S. aureus* (MSSA) and methicillin-resistant *S. aureus* (MRSA)) and aerobic Gram-negative bacilli (e.g., *Pseudomonas aeruginosa*, *Escherichia coli*, *Klebsiella pneumoniae*, *Enterobacter* spp., *Acinetobacter* spp.). The patterns of the causative pathogens in NP are affected by host factors, including bacterial colonization of the stomach and oropharynx and hospital flora [1,2,3]. Both bacteria of the host and hospital flora could be represented by pathogens resistant to commonly used antimicrobials (the phenomenon of antimicrobial resistance, AMR). The ongoing spread of AMR, which is now recognized one of the leading threats to public health in the 21st century [4], has made pneumonia treatment increasingly difficult. The problem has been further exacerbated by the SARS-CoV2 pandemic, as a severe overload of the healthcare system and especially intensive care units (ICUs) may have contributed to selection and spread of drug-resistant pathogens, primarily MRSA, carbapenem-resistant *Acinetobacter*
*baumannii* and *Klebsiella pneumoniae* and the fungus *Candida auris* [5].

The estimated incidence of community-acquired pneumonia worldwide ranges from 1.5 to 14 cases per 1000 person-years, depending on geographical, seasonal and demographic characteristics [6]. The frequency of HAP ranges from 5 to more than 20 cases per 1000 hospital admissions, being the second most common nosocomial infection after urinary tract infection and the leading cause of death from nosocomial infections in critically ill patients [3,6,7]. One of the frequent and serious pulmonary conditions co-occurring with severe pneumonia is acute respiratory distress syndrome (ARDS), which is defined as the acute onset of non-cardiogenic pulmonary edema, hypoxemia and the need for mechanical ventilation [8]. Overall, 10% of all ICU patients and 23% of mechanically ventilated patients may suffer from ARDS [9]. In different studies involving patients with ARDS of any etiology, mortality varied in a wide range from 11% to 87% [10]. Severe pneumonia can co-occur with sepsis, which is defined as a life-threatening organ dysfunction caused by a dysregulated host response to infection [11]. Mortality rates from sepsis range from 15% to 56% [12]. Both ARDS and sepsis can develop as a result of pneumonia, or precede it [1,13].

Approximately 450 million people a year suffer from pneumonia globally. Chronic comorbidities and age affect the development and outcome of pneumonia [1]. The highest rates of morbidity and mortality from pneumonia are observed in children under the age of 5 and adults over 70 years of age. In the literature, pneumonia statistics are usually analyzed in detail for children under the age of 15–16 and the elderly, and a large group of young and middle-aged adults is considered together, which implies relatively constant data for this long period [14]. However, morbidity, hospitalization and mortality due to pneumonia also increase with age in this group [15,16,17,18]. The decline of the adaptive immune system with age has been recognized as the main risk factor for some viral and bacterial diseases caused by SARS-CoV-2, MERS-CoV, West Nile virus, Influenza A virus, MRSA, group A and group B streptococcus, Legionella, *Streptococcus pneumoniae* and *Haemophilus influenzae* [19]. With COVID-19, the likelihood of hospitalization exponentially depends on age, doubling every 16 years, which corresponds to an increase of 4.5% for every year of life. This reflects an exponential decline in both thymus volume and T-cell production, which are halved every 16 years [19,20].

The adaptive immune system is considered to be very specific [21]. Its major factors are T cells and B cells derived from the multipotent hematopoietic stem cells of the bone marrow. After formation in the bone marrow, progenitor T cells migrate to the thymus to mature and turn into T cells, while B cells remain to mature in the bone marrow. During maturation, the precursors of T and B cells undergo V(D)J-recombination with the formation of diverse and functional repertoires of TCR and BCR (T- and B-cell receptors), which allow the recognition of an almost unlimited number of antigens [22,23]. The processes of TCR and BCR rearrangement are accompanied by the formation of circular DNA elements, T-cell receptor excision circles (TRECs) and kappa-deleting recombination excision circles (KRECs). TRECs and KRECs are nonreplicating pieces of DNA persisting in the cells. Since TRECs and KRECs are unable to replicate, diluting after each cell division, they are considered markers of new lymphocyte output. TRECs, in particular, are enriched in recent thymic emigrants (RTE) [23,24]. The TREC/KREC assessment for profiling thymic and bone marrow production is widely used in newborn screening programs for severe combined immunodeficiency [25]. TREC assays have shown their usefulness in clinical settings in which T-cell immunity is important, including the diagnosis and monitoring of T-cell immunodeficiency, HIV infection, aging, autoimmune diseases and immune reconstitution after bone marrow transplantation [26]. We and others have recently shown that reduced levels of TRECs and KRECs in adult patients with COVID-19 correlate with the development of ARDS, disease progression and unfavorable outcomes [27,28]. It can be expected that TREC and KREC assays will demonstrate their diagnostic ability for other infectious diseases as well.

Pneumonia’s development, severity and outcome depend on age, comorbidities and the host immune response [1]. Our work is aimed at exploring the relationships between these conditions. We combined theoretical and experimental studies to characterize pneumonia with its comorbidities and age distributions and to evaluate the diagnostic utility of TRECs/KRECs as markers of immune response in patients with CAP and NP. Evidence for the association of comorbidities with pneumonia is based on studies generally focused on specific classes of diseases and specific age groups [1,15,16,18]. These data often vary greatly and do not provide a holistic view of the entire spectrum of pneumonia-related conditions. Optimal knowledge acquisition can be achieved using open access data aggregating information for large cohorts of patients. An example of such a resource is the Columbia Open Health Data (COHD) resource, which provides access to clinical concept prevalence and co-occurrence from electronic health records (EHR) [29]. The theoretical part of the work was performed using COHD. We focused on pneumonia co-occurrence with sepsis and ARDS in comparison with other comorbidities and then expanded the analysis to pneumonia caused by various pathogens. We also examined age-specific distribution of pneumonia, sepsis and ARDS. The experimental part of the work included the measurement of TREC/KREC values in patients with CAP or NP. In order to reduce the impact of covariates such as age and comorbid diseases, only young patients (18–40 years of age) without advanced chronic diseases were selected to study the diagnostic capabilities of TREC and KREC assays in pneumonia development and outcome.

## 2. Results

### 2.1. Theoretical Study: Pneumonia and Its Acute and Chronic Comorbidities in the COHD Database

#### 2.1.1. Overview of Pneumonia, Sepsis and ARDS and Co-Occurring Conditions

In the 5-year non-hierarchical COHD dataset (1,790,431 patients), the patient counts in pneumonia, sepsis and ARDS were 31,783, 16,269 and 1143, respectively (Figure 1A). In all three sets, the largest numbers of co-occurring conditions classified according to ICD-10 were represented by diseases of the circulatory system. Diseases of the circulatory system were followed by neoplasms in pneumonia, diseases of the digestive system in sepsis and symptoms, signs and abnormal clinical and laboratory findings not elsewhere classified in ARDS (Figure 1A). For pneumonia, the top 15 co-occurring items within the ICD-10 top-level categories are given in Appendix A. Among diseases of the circulatory system, congestive heart failure was most prevalent. Other types of acute and chronic heart failure were also among top 15 terms. The most significant associated neoplasms included, in particular, respiratory and blood malignancies.

Since each concept can be described by several ICD-10 codes, the total number of co-occurring conditions for the diseases under consideration was less than the number of ICD-10 codes (Figure 1B). Due to overlapping, only 22.01%, 4.61% and 10.38% of concepts were unique in pneumonia, sepsis and ARDS, respectively. Figure 1C shows O/E and RF for top 15 co-occurring conditions in patients with pneumonia, sepsis and ARDS. The results of the concept association analysis showed that the strength of the dependence between pneumonia and sepsis (O/E) was 25.1, and the frequency of pneumonia among patients with sepsis (RF) was 44.6%, while the frequency of sepsis in patients with pneumonia was 22.8%. ARDS occurred in 24.8% of patients with pneumonia (O/E is 14.0) and in 29.1% of patients with sepsis (O/E is 16.4). With the concepts of interest, the top co-occurring terms included acute respiratory and renal failure and sepsis-related critical conditions typical for ICUs [1,13].

#### 2.1.2. Temporal Associations of Pneumonia, Sepsis and ARDS

Temporal analysis was based on clinical data from the life-time dataset for pneumonia (patient count 121,241), sepsis (patient count 49,273) and ARDS (patient count 1171). The age distribution for two pairs of concepts, pneumonia/sepsis and pneumonia/ARDS is shown in Figure 2. Children in the first years of life represented high (pneumonia) or extremely high (sepsis and ARDS) percentage of patients with these conditions. Between 10 and 20 years of age, the number of patients became minimal. A gradual increase in age-related prevalence of pneumonia and sepsis began approximately at the age of 20 and lasted until the age of 65–70. A dramatic increase in the number of patients with pneumonia and sepsis was seen in the 90+ age group.

#### 2.1.3. Pathogen-Specific Pneumonia and Its Co-Occurrence with Sepsis, ARDS and Disorders of Immune Function

From the total pool of pneumonia-related concepts in the COHD 5-year non-hierarchical dataset, we selected 13 pathogen-specific concepts with a patient count ≥ 100 (Figure 3A). The largest sample of viral pneumonia was represented by pneumonia caused by respiratory syncytial virus (547 patients, 507 associated concepts). In bacterial pneumonia, the largest samples of Gram-positive and Gram-negative bacteria were, respectively, represented by staphylococcal pneumonia (704 patients, 754 concepts) and pneumonia caused by *Pseudomonas* (1018 patients, 892 concepts). The number of concepts correlated with the patient counts (*r* = 0.86). Pneumonia caused by *Pseudomonas* and staphylococcal pneumonia were found to co-occur most often with pneumonia caused by other pathogens including viruses (Figure 3B). With one exception, sepsis and ARDS were among the co-occurring conditions in all sets of pathogen-specific pneumonia (Figure 3C). Sepsis was characterized by higher levels of O/E and RF in bacterial rather than in viral pneumonia, while in ARDS these values were similar or higher in viral than in bacterial pneumonia. We also found that disorder of immune function was associated with all types of pathogen-specific pneumonia (Figure 3C).

### 2.2. Experimental Study: Host Immune Response Measured Using TRECs/KRECs in Patients with Community-Acquired and Nosocomial Pneumonia

#### 2.2.1. Demographic and Clinical Data

Among the 433 participants included in the study, 263 had CAP or NP. The control group consisted of 170 people (77.6% males) aged 23.52 ± 5.93 years. Data on smoking status were available for 123 persons, and among them there were 71 current smokers. Controls were older than CAP patients (*p* = 9.56 × 10^−4^) and younger than NP patients (*p* = 7.0 × 10^−6^). CAP patients included more males (*p* = 5.01 × 10^−8^) and more smokers than controls (*p* = 3.41 × 10^−6^).

The characteristics of the CAP and NP groups are presented in Table 1.

Microbiological confirmation was obtained in 69.5% of CAP patients and in 45.5% of NP patients. *Streptococcus* species were isolated in 79 patients (69.3% of patients with a known causative microorganism), and *Streptococcus pneumoniae* was the most common etiologic agent (32 patients, 28.1%). *Staphylococcus aureus* and *Haemophilus influenzae* were identified in 27 and 24 patients with CAP, respectively (23.7% and 21.1%). Typical nosocomial pathogens known to play a role in the development of NP, including VAP, were identified in 46 patients with NP for whom microbiological data were available and included Gram-negative bacilli (35 patients, 76.1% of the patients with a known causative microorganism), Gram-positive bacteria (3 patients, 6.5%) and mixed cases (8 patients, 17.43%). *Pseudomonas aeruginosa* (15 patients, 32.6%), *Klebsiella pneumoniae* (11 patients, 23.9%) and *Acinetobacter baumannii* (11 patients, 23.9%) were more common than other pathogens in NP. The clinical data for patients with CAP and NP differed greatly. Compared to patients with CAP, patients with NP had more severe pneumonia measured using PSI (*p*-value for likelihood ratio χ^2^ 2.4 × 10^−14^), they were more likely to undergo mechanical ventilation, stayed longer in the hospital and ICU and more often had adverse outcomes.

#### 2.2.2. TREC and KREC Levels in Healthy Controls, CAP and NP Patients

TREC and KREC levels did not differ between the age groups < 30 years and ≥30 years and between males and females in the control group (Appendix A). Smoking did not affect TREC and KREC concentrations in healthy controls, CAP and NP patients with one unexpected exception for higher KREC values in smokers with NP (*p* = 0.02); however, these results remained marginally significant after correction for two comparisons (Appendix A). TREC levels in controls were lower than in CAP patients and higher than in NP patients (Figure 4A). KREC levels were higher in controls than in NP patients and did not differ between controls and CAP patients. Analysis of TRECs/KRECs in the groups of patients stratified by pathogens (CAP: *Streptococcus pneumoniae*, *Staphylococcus aureus* and *Haemophilus influenzae*; NP: *Pseudomonas aeruginosa*, *Klebsiella pneumoniae* and *Acinetobacter baumannii*) did not reveal differences in the distribution of TRECs/KRECs depending on the pathogen (Appendix A). A significant inverse correlation between TREC levels and age was evident in the CAP and NP groups, but it was weak in the control group (Figure 4B). All patients with TREC levels corresponding to the 10th decile in the CAP (>901.1 copies/10^5^ cells) (*n* = 16) and NP groups (>291.4 copies/10^5^ cells) (*n* = 10) were younger than 30 years (Figure 4B). KREC levels did not significantly decrease with age in any of the groups (Figure 4B).

#### 2.2.3. TREC and KREC Levels in Pneumonia with Varying Severity and Outcome

TREC and KREC levels were lower in patients with more severe pneumonia according to PSI as well as in patients with bilateral compared to unilateral pneumonia (Figure 5A,B). There was a partial overlap between patients with ARDS, sepsis/septic shock and lethal outcomes (Figure 5C). All three phenotypes considered were associated with lower TREC levels, with the strongest association found in deceased versus discharged patients (Figure 5D). The same effect was observed for KREC levels, although the significance of the effects was lower.

#### 2.2.4. Diagnostic Accuracy of TREC and KREC Assays

Based on the rough classifying system (Safari et al., 2016), the TREC assay demonstrated fair diagnostic performance for NP, ARDS, and sepsis/septic shock (AUC 0.7–0.8) and good diagnostic performance (AUC 0.8–0.9) for lethal outcome (Figure 6). Compared to the TREC assay, the KREC assay had worse diagnostic accuracy in relation to all the studied endpoints (Figure 5, Appendix A).

## 3. Discussion

In this study, we provided data from the COHD 5-year dataset on the relationship between pneumonia, sepsis and ARDS in terms of their O/E ratios and RFs. Co-occurring rates between these and some other closely related conditions common in ICUs were the top among 4170, 3382 and 963 comorbidities in pneumonia, sepsis and ARDS, respectively. In pathogen-specific types of pneumonia, these associations also were observed. Associations were higher in pairs of diseases associated with bacterial pneumonia/sepsis and viral pneumonia/ARDS. Using the COHD life-time dataset with the patient counts 121,241, 49,273 and 1171 for pneumonia, sepsis and ARDS, respectively, we showed that in adults, the incidence of pneumonia and sepsis gradually increased starting from about 20 years to 65–70 years, while age-dependent ARDS data were less consistent. Given the role of adaptive immunity in the manifestation and progression of infectious disease, as well as the decline in T-cell immunity after the juvenile period [19], we suggested that measurements of thymus and bone marrow production using TREC and KREC assays may be predictive for the development and outcome of pneumonia. To check this hypothesis, we conducted a study of TREC and KREC levels in young subjects (18–40 years of age) with CAP, NP and healthy controls and compared these markers in patients with different severity levels of pneumonia and outcomes. We found that low TREC and to a lesser extent KREC levels were associated with the risk of an adverse outcome in these groups of patients.

In the COHD 5-year dataset, in addition to the high rates of co-occurrence between pneumonia, sepsis and ARDS, we revealed that these conditions were associated with a wide range of comorbidities from all anatomical or etiological groups in the ICD-10 classification system. The greatest number of co-occurring conditions was represented by diseases of the circulatory system, which is consistent with the literature data on numerous bidirectional causal relationships between pneumonia and cardiovascular disease [30]. Diseases classified as neoplasms ranked second in the list of most common conditions associated with pneumonia. The link between pneumonia and neoplasms may be mediated by age-related deterioration of immune function as the main cause of both cancer and pneumonia [19,31,32].

Our results of the experimental part of the study indicate the absence of pathogen-specific differences in the distribution of TRECs and KRECs. This observation seems reasonable since the development of pneumonia depends more on the host immune response than on the characteristics of the pathogen [1]. Genetics underlies the immune profile, and the number of people potentially susceptible to infectious diseases may be underestimated [33]. We showed a high diversity of TREC and KREC levels in all groups. Patients with CAP had the increased levels of TRECs compared to controls, which probably reflects the protective production of the thymus as an early response to infection [34]. These data are consistent with the findings of Cuvelier et al., who showed the protective reactive thymus hyperplasia and increased TRECs as a proxy for thymic output in adult patients with COVID-19 [35]. Patients with thymus hyperplasia had a lower mortality rate compared to patients lacking thymic activity/reactivation. The protective reaction depended on age, which also correlates with our data.

In patients with severe illness, increased lymphocyte apoptosis and redistribution of lymphocytes to other tissues than peripheral blood can contribute to the identified TREC and KREC counts [36,37]. In our study, TREK and KREC levels were significantly reduced in patients with NP compared to the control group, which is partly explained by the underlying conditions resulting in NP. Both severe pneumonia and critical illness were associated with low TREC and KREC levels. Taking into account the prospective character of our study in relation to the development of NP and critical conditions and the results of the ROC analysis, we believe that measurements of TREC and KREC levels on admission have predictive value in patients with pneumonia.

The results of the study may be of interest in connection with the AMR problem. In immunodeficient patients, a personal history of frequent infections requiring the administration of broad-spectrum antibiotics seems to be associated with AMR [38]. Given these data, individuals with suboptimal levels of TRECs/KRECs and, consequently, insufficient immune response may represent high-risk AMR groups. To reduce the spread of AMR, rapid diagnosis and patient management based on knowledge of the main combinations of pathogens and drugs that contribute to the development of AMR in each specific medical facility are crucial. Monitoring access to essential antibiotics, as well as research and development of new vaccines and antibiotics, are also important [4,5]. The use of TREC/KREC measurements in clinical practice can contribute to the mitigation of the problem. Patients with reduced levels of TRECs/KRECs identified upon admission need individual support to prevent AMR and increase the likelihood of a favorable outcome.

The limitations of the theoretical analysis are the same as in the COHD study [29]. In the context of our interest, the main limitation is that the co-occurrence data did not comprise temporal relationships between concepts. The main limitations of the experimental study include different age and sex characteristics of patients and controls, different timing of TREC and KREC measurements and the absence of other hematological and immunological data. Due to the high frequency of mixed infections, the comparison groups for assessing the pathogen-specific distribution of TRECs/KRECs included both patients with single and mixed infections.

## 4. Materials and Methods

### 4.1. Theoretical Study with Columbia Open Health Data (COHD)

A publicly accessible database, COHD, derived from Columbia University Irving Medical Center’s Observational Health Data Sciences and Informatics (OHDSI) database, which includes inpatient and outpatient data, contains information on prevalence and co-occurrence of medical conditions from EHR. Several datasets are available, including a lifetime non-hierarchical dataset with clinical data for 1985–2018 and a 5-year non-hierarchical dataset that covers the period with the most stable clinical data for 2013–2017 [29]. The lifetime dataset includes 36,578 single concepts (conditions, drugs and procedures, i.e., clinical entities and events) and 32,788,901 concept pairs from 5,364,781 patients. The 5-year dataset contains 29,964 single concepts and 15,927,195 concept pairs from 1,790,431 patients. The resource provides direct access to pairs of co-occurring diseases and the results of the analysis of associations between pairs of diseases (https://cohd.io/index.html) (accessed on 1 July 2022). Association methods include χ^2^, observed-to-expected frequency ratio (O/E) and relative frequency (O/E). The χ^2^ and *p*-value (initial and Bonferroni-adjusted) between pairs of concepts show the relationship between two concepts. O/E (represented as a natural logarithm) demonstrates the ratio between the observed count and the expected count for a pair of concepts. The relative frequency indicates how often the C1 concept (a concept of interest) occurs in patients who have a C2 concept (a co-occurring concept).

Co-occurring conditions (domain ‘condition’) for the concepts pneumonia (concept ID: 255848), ARDS (concept ID: 4195694) and sepsis (concept ID: 132797) as well as concepts related to pathogen-specific pneumonia (*n* = 13) (Appendix A) and disorder of immune function (concept ID: 440371) were selected using a Bonferroni-adjusted *p*-value < 0.05 for the χ^2^ test. The natural logarithm of the O/E ratio was turned back to the original scale. The concept names given according the SNOMED vocabulary (https://snomedbrowser.com/) (accessed on 2 July 2022) were converted to ICD-10 codes. Data processing was implemented using COHD RESTful SmartAPI (https://github.com/WengLab-InformaticsResearch/cohd_api/blob/master/notebooks/COHD_API_Example.ipynb) (accessed on 20 July 2022).

### 4.2. Experimental Study

#### 4.2.1. Patients and Controls

Young patients (18–40 years of age) with CAP and NP were recruited from those hospitalized at clinical hospitals of the Federal Research and Clinical Center of Intensive Care Medicine and Rehabilitology, Moscow, Russia from January 2008 to December 2019 [39]. Exclusion criteria for the CAP and NP groups included lack of informed consent, conditions related to the immune system (infections, anaphylactic conditions, autoimmune diseases), chronic respiratory diseases, malignancy, decompensated chronic conditions, extensive bleeding and pregnancy. The control group consisted of age-matched individuals who claimed that at the time of participation in the study they had no chronic or acute diseases. The diagnosis of CAP was based on the presence of acute symptoms of lower respiratory tract infection confirmed by clinical data (cough, sputum discharge, auscultation data corresponding to pneumonia, temperature > 38 °C or <35 °C), X-ray data (infiltrate on chest X-ray) and laboratory (leukocyte count > 10 × 10^9^ cells/L or <4 × 10^9^ cells/L) data. The NP group consisted of accident victims with physical trauma and patients requiring surgery who developed NP during their stay in the hospital. The diagnosis of NP was made based on a new infiltration on a chest X-ray in the presence of clinical and laboratory data. To ensure blood sampling at admission, clinical data for the NP group were collected prospectively.

The severity of patient conditions was assessed using Acute Physiology and Chronic Health Evaluation (APACHE) II scores [40], which were calculated within 24 h after patient admission. The diagnosis of sepsis has been revised in accordance with the 2021 version of the Surviving Sepsis Campaign guidelines [11]. The diagnosis of ARDS was based on the Berlin definition [41]. The severity of pneumonia was assessed using the Pneumonia Severity Index (PSI) [42]. CAP and NP treatment decisions were based on current versions of national guidelines [43,44].

Respiratory fluid samples obtained by endotracheal aspiration were taken for microbiological documentation of CAP and NP. The bronchoalveolar lavage (BAL) fluid was stained and cultured for aerobic and anaerobic bacteria. A positive quantitative culture was defined when bacteria were cultured from BAL samples at a concentration of ≥1 × 10^5^ CFU/mL.

The study protocol was approved by the institutional review board (IRB) at the Federal Research and Clinical Center of Intensive Care Medicine and Rehabilitology. Written informed consent was signed by all patients included in the study. The study was conducted under the tenets of the Declaration of Helsinki.

#### 4.2.2. Generation of Control Constructs for TRECs and KRECs

Human albumin, the δREC–ψJα T cell receptor, and the kappa-deleting joint were PCR amplified from DNA obtained from the peripheral blood of a healthy donor and separately cloned into the pBlueScript SK(II-) vector (Addgene). The TREC fragment was ligated to the SpeI restriction site, the KREC kappa deletion joint was cloned in HindIII and SpeI restriction sites. DNA plasmids were transformed into *E. coli.* Efficiency was tested on a series of dilutions of the construct.

#### 4.2.3. Real-Time Quantitative PCR (RQ-PCR) for TRECs and KRECs

Blood samples taken at patient admission were used to measure TREC/KREC levels. DNA was isolated from peripheral blood by protein precipitation followed by DNA extraction with isopropanol. Primers and probes were designed for specific amplification of δREC–ψJα T-cell receptor, kappa-deleting joint and human albumin (reference gene). The 20 mL qPCR mixture contained 1.5 µM primers for sj-TREC, 3 µM primers for KREC, 1.5 µM primers for the albumin gene, 0.25 µM FAM-labeled TaqMan probes for sj-TREC and Cy5-labeled TaqMan probes for the albumin gene, 0.5 µM HEX-labeled TaqMan probes for KREC, 1 × buffer for HS Taq DNA polymerase (Eurogen, Moscow, Russia), 0.2 mM each of dNTP, 5 mM MgCl_2_, 0.5 units HS Taq DNA polymerase (Eurogen, Moscow, Russia), 150 ng of DNA and 0.4 ng BSA. qPCR was performed in a CFX96 thermocycler (Bio-Rad, Hercules, CA, USA). Albumin amplification was performed in the same plate. The sequence of primers and probes is given in Appendix A. The standard protocol consisted of initial heating at 95 °C for 7 min, followed by 45 denaturation cycles at 93 °C for 30 s and combined primer/probe annealing and elongation at 61.7 °C for 1 min. TRECs, KRECs and copies of the albumin gene were calculated by extrapolating the corresponding amounts of the sample from the standard curve obtained by serial dilutions (10^5^, 10^4^, 10^3^, 10^2^ and 10 (50 for KRECs)) of the plasmid DNA, containing δREC–ψJα T-cell receptor, kappa-deleting joint, human albumin, which were amplified on the same PCR plate. The number of TRECs/KRECs per 100,000 cells was calculated using the following formula:$$\frac{mean\;TREC\left(KREC\right)value}{\frac{mean\;albumin\;value}{2}}\times 100,000$$

The detection limit for quantitative PCR was approximately 10 TREC or 50 KREC copies. The results were considered acceptable if the standard curve slopes ranged from −3.55 to −3.30, the PCR efficiency ranged from 91% to 110%, and the correlation coefficients were higher than 0.99. Samples with an albumin level less than 20,000 copies per reaction (*n* = 14) were excluded.

#### 4.2.4. Statistical Analysis

The calculations were carried out using the R (version 3.4.1) software. For categorical variables, the exact two-tailed Fisher test or the likelihood ratio χ^2^ test for two or more independent comparison groups, respectively, were used to assess whether the proportion of the studied variables was the same in the compared samples. For continuous variables, we first checked the normality of the distribution using the Shapiro–Wilk test, which showed a significant deviation from normality for most samples. Therefore, to assess whether two compared samples were pooled from the same sample, we used the Mann–Whitney U test. The Kruskal–Wallis test was used to compare more than two groups. Continuous variables were represented as mean ± SD. To measure the strength and direction of the relationship between two variables, we used Spearman’s non-parametric correlation coefficient. The significance level at which the null hypothesis of the absence of differences between the study groups was rejected was set at 0.05. The Bonferroni correction was used to correct for multiple comparisons separately for demographic indicators (controls, CAP and NP groups) and clinical conditions (CAP and NP groups). The calculation of the area under receiver operating characteristic (ROC) curve (AUC) was performed using the approach of DeLong et al. [45]. The accuracy of TREC and KREC measurements was evaluated in patients with NP compared with healthy controls and in patients with ARDS, sepsis and lethal outcome compared with patients without ARDS, sepsis and lethal outcome.

## 5. Conclusions

We characterized pneumonia, sepsis and ARDS in terms of their comorbidities and demonstrated a gradual increase in the incidence of pneumonia and sepsis beginning in adults at the age of approximately 20 years. We also showed that in young adults with CAP, high TRECs are common and could be associated with a favorable outcome due to the activation of protective mechanisms. In patients with pneumonia, low TREC and KREC levels are potential predictors of severe disease and an adverse outcome. Further characterization of TREC and KREC age-specific reference intervals in healthy individuals is required for using these parameters as diagnostic tools in adults. Studying the correlation of TRECs/KRECs with AMR may become a future research direction with significant scientific and clinical impact.

## Figures and Tables

**Figure 1 ijms-23-09896-f001:**
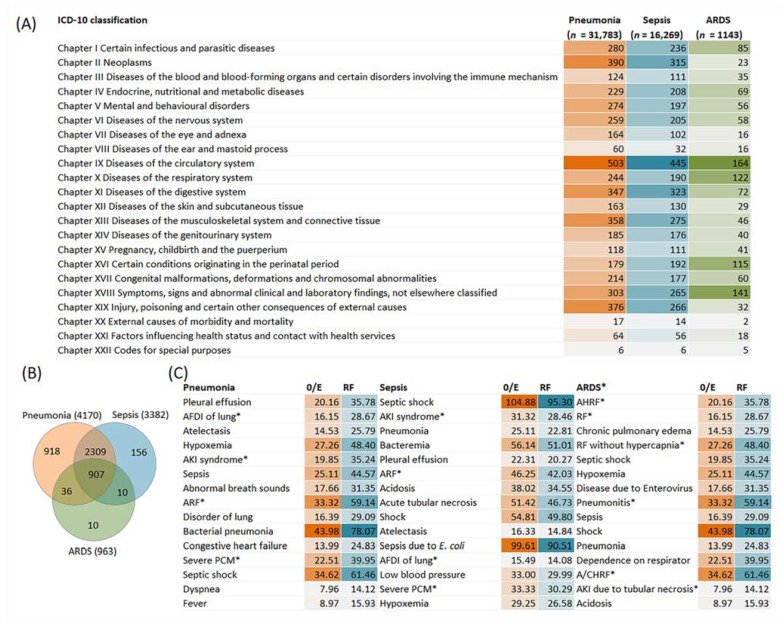
Summary of COHD data from the 5-year non-hierarchical dataset for clinical conditions co-occurring with pneumonia, sepsis and acute respiratory distress syndrome (ARDS). (**A**) The number of ICD-10 classifications for all significant associations according the range of codes in each chapter. (**B**) Venn diagram for co-occurring concepts sharing by pneumonia, sepsis and ARDS. Since each concept can be described by several ICD-10 codes, the total number of co-occurring conditions is less than the number of ICD-10 codes. (**C**) Observed-to-expected frequency ratios (O/E) and relative frequencies (RF) for the top 15 co-occurring conditions in accordance with the results of the χ^2^ test. The natural logarithm of the O/E ratio has been converted back to the original scale. * Abbreviations: AKI, acute renal (kidney) failure; (A)RF, (acute) respiratory failure; A/CHRF, acute on chronic hypoxemic respiratory failure; AFDI, abnormal findings on diagnostic imaging of lung; AHRF, acute hypoxemic respiratory failure; PCM, protein–calorie malnutrition; Pneumonitis, pneumonitis due to inhalation of food or vomitus.

**Figure 2 ijms-23-09896-f002:**
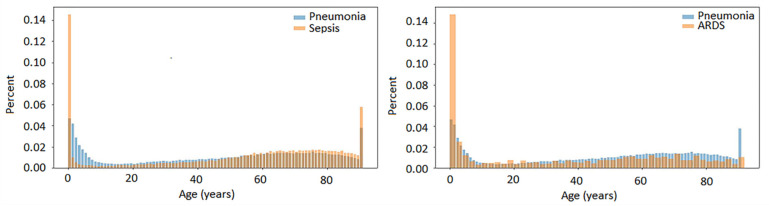
Temporal clinical data for pneumonia, sepsis and ARDS. Concept-age distributions.

**Figure 3 ijms-23-09896-f003:**
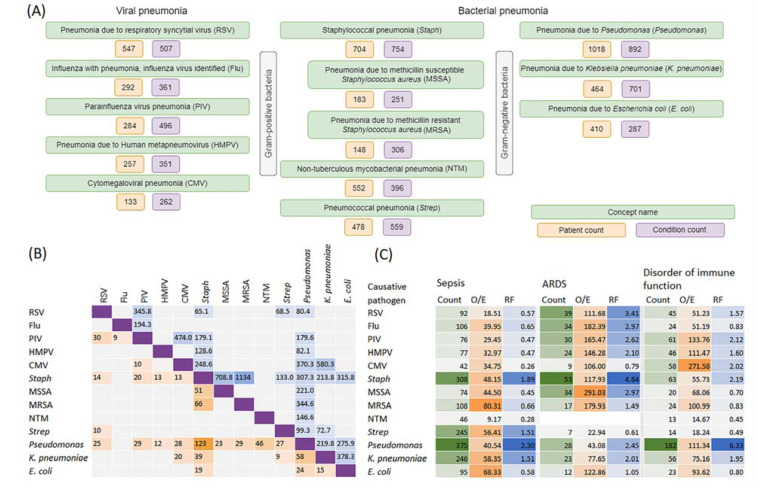
Pathogen-specific pneumonia concepts in the 5-year non-hierarchical COHD dataset. (**A**) Summary data reflecting the patient and co-occurring condition count for 13 pathogen-specific types of pneumonia. (**B**) Heat-map matrix for co-occurring types of pathogen-specific pneumonia: observed-to-expected frequency ratios (O/E) (above the purple diagonal divider) and the patient count in co-occurring conditions (below the purple diagonal divider). (**C**) O/E and relative frequencies (RF) of pathogen-specific pneumonia in patients with sepsis, acute respiratory distress syndrome (ARDS) and disorder of immune function. The natural logarithm of the O/E ratio has been converted back to the original scale.

**Figure 4 ijms-23-09896-f004:**
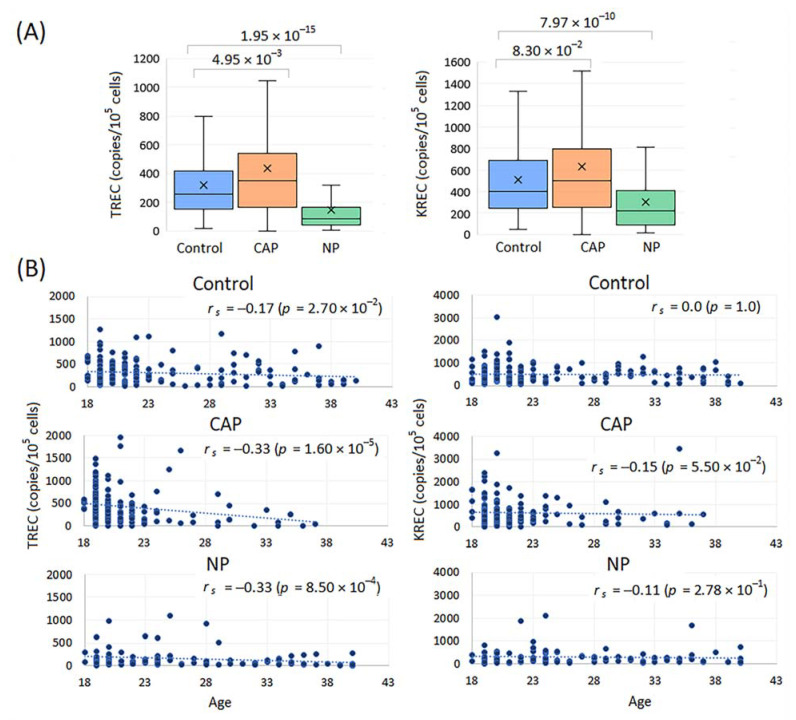
TREC/KREC counts in healthy controls and patients with CAP and NP. (**A**) Boxplots depicting differences between TREC/KREC levels in healthy controls and patients with community-acquired (CAP) and nosocomial (NP) pneumonia. (**B**) Scatterplots showing the correlation between age and TREC/KREC levels in healthy controls and patients with CAP and NP. Spearman’s rank correlation coefficient *r_s_* and associated *p*-value (one tail) are indicated.

**Figure 5 ijms-23-09896-f005:**
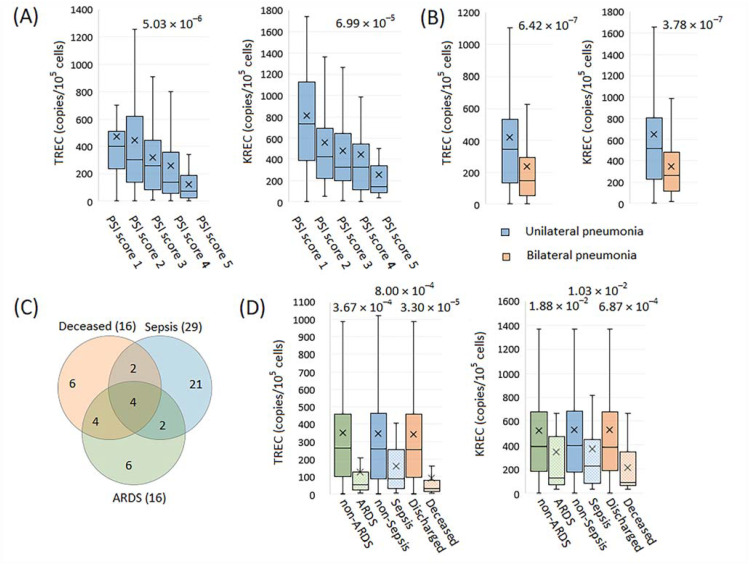
TREC/KREC counts related to pneumonia severity and outcome. Community-acquired and nosocomial pneumonia are considered together. Box-plots depicting differences between TREC/KREC levels in patients (**A**) with different Pneumonia Severity Index (PSI) and (**B**) with bilateral compared to unilateral pneumonia. (**C**) Venn diagram of acute respiratory distress syndrome (ARDS), sepsis and lethal outcome in patients with pneumonia. (**D**) Box-plots for TREC/KREC levels in patients with ARDS, sepsis and lethal outcome versus patients without ARDS, sepsis and lethal outcome (discharged). (**C**,**D**) The sepsis group included patients with sepsis and septic shock.

**Figure 6 ijms-23-09896-f006:**
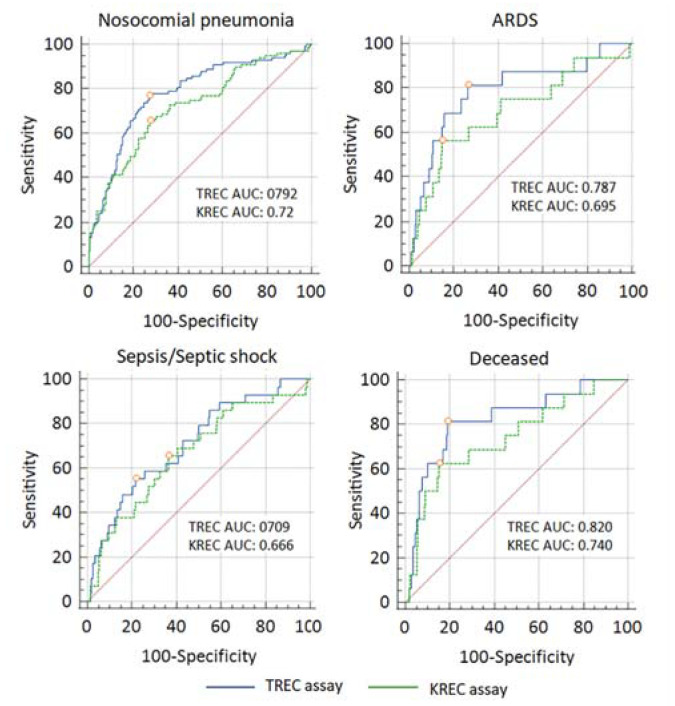
Receiver operating characteristic (ROC) curves for TREC/KREC levels in predicting the development, progression and outcome of pneumonia. The red dots on the ROC curves indicate the position of the optimal cut-offs determined by Yudin’s J-statistics. The red diagonal line denotes the ROC curve of a random classifier. Other summary statistics on the results of the ROC analyses are presented in Appendix A.

**Table 1 ijms-23-09896-t001:** Characteristics of the CAP and NP groups.

Feature	CAP (%)	NP (%)	*p*
Total number	164	99	
Age	21.29 ± 3.89	26.68 ± 6.68	4.52 × 10^−14^
Sex ratio (M)	159 (96.9)	93 (93.9)	0.34
**Current smoking status**			
Yes	126	44	0.09
No	25	17	-
No data	13	38	-
**Pre-existing conditions**			
Cardiovascular diseases	0 (0.0)	2 (2.0)	0.14
Gastric/duodenal ulcer	1 (0.6)	2 (2.0)	0.56
Neurological conditions	2 (1.2)	1 (1.0)	1.0
Obesity	1 (0.6)	3 (3.0)	0.15
Musculoskeletal disorders	1 (0.6)	2 (2.0)	0.56
Benign neoplasms	0 (0.0)	1 (1.0)	0.38
Genitourinary system diseases	1 (0.6)	2 (2.0)	0.56
**Initial diagnosis**			
Trauma	0 (0.0)	90 (90.9)	-
Acute poisoning	0 (0.0)	2 (2.0)	-
Surgery	0 (0.0)	7 (7.1)	-
CAP	164 (100)	0 (0.0)	-
**Disease history**			
Day of CAP on admission	8.01 ± 7.48	-	
Day of NP development	-	4.93 ± 2.48	
**General information on infectious pathogens in BAL fluid**			
Microbiological data	114 (69.5)	46 (45.5)	2.62 × 10^−4^
Gram-positive bacilli (GPB)	72 (63.2)	3 (6.5)	5.51 × 10^−9^
Gram-negative bacilli (GNB)	1 (0.9)	35 (76.1)	1.83 × 10^−24^
Mixed GPB + GNB	41 (36.0)	8 (17.4)	0.023
Single infection	48 (42.1)	6 (13.0)	3.84 × 10^−4^
**Main pathogens in BAL fluid**			
*Streptococcus pneumoniae*	32 (28.1)	2 (4.3)	5.09 × 10^−4^
*Streptococcus pyogenes*	30 (26.3)	2 (4.3)	9.64 × 10^−4^
*Streptococcus* spp.	17 (14.9)	1 (2.2)	0.025
*Staphylococcus aureus* *➢ MRSA*	27 (23.7) ➢ 0 (0.0)	6 (13.0) ➢ 5 (10.9)	0.19 ➢ 1.70 × 10^−^^3^
*Haemophilus influenzae*	24 (21.1)	5 (10.9)	0.17
*Enterobacter* spp.	6 (4.2)	3 (6.5)	0.72
*Pseudomonas aeruginosa*	1 (0.9)	15 (32.6)	1.38 × 10^−8^
*Klebsiella pneumoniae*	7 (6.1)	11 (23.9)	3.90 × 10^−3^
*Escherichia coli*	9 (7.9)	8 (17.4)	0.09
*Acinetobacter baumannii*	6 (4.2)	11 (23.9)	1.18 × 10^−3^
*Other*	5 (4.4)	7 (15.2)	0.04
**Clinical data**			
APACHE-II	9.37 ± 5.13	16.70 ± 5.55	2.85 × 10^−19^
PSI score 1	34 (20.7)	0 (0.0)	3.85 × 10^−8^
PSI score 2	42 (25.6)	6 (6.1)	5.42 × 10^−5^
PSI score 3	60 (36.6)	47 (47.5)	0.09
PSI score 4	24 (14.6)	32 (32.3)	1.0 × 10^−3^
PSI score 5	4 (2.4)	14 (14.1)	5.48 × 10^−4^
Bilateral	47 (28.7)	73 (73.7)	8.44 × 10^−9^
Mechanical ventilation	10 (6.1)	35 (35.4)	3.47 × 10^−9^
Duration of mechanical ventilation, days	1.3 ± 0.64	10.03 ± 4.82	6.0 × 10^−6^
ICU admission	31 (18.9)	85 (85.9)	3.11 × 10^−9^
ICU length of stay, days	4.10 ± 2.75	18.28 ± 14.35	2.40 × 10^−9^
Hospital length of stay, days	14.01 ± 6.62	48.30 ± 19.59	3.0 × 10^−39^
ARDS	2 (1.2)	14 (14.1)	3.45 × 10^−5^
Sepsis/septic shock	0 (0.0)	29 (29.3)	2.56 × 10^−14^
Deceased	2 (1.2)	14 (14.1)	3.50 × 10^−5^

Due to Bonferroni’s correction, significance was reached when *p* < 0.005 for demographics data and <0.001 for clinical data. The significance of microbiological data was calculated for a subgroup of patients with identified pathogens. Abbreviations: APACHE-II, Acute Physiology and Chronic Health Evaluation II scores; ARDS, acute respiratory distress syndrome; CAP, community-acquired pneumonia; BAL, bronchoalveolar lavage; ICU, intensive care unit; MRSA, methicillin-resistant *S. aureus*; NP; nosocomial pneumonia; PSI, Pneumonia Severity Index.

## Data Availability

The data presented in this study are available on request from the corresponding author.

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
