# Peer review of "Pneumonia and Related Conditions in Critically Ill Patients—Insights from Basic and Experimental Studies"

_ijms, 2022, doi:10.3390/ijms23179896_

Round 1

Reviewer 1 Report

General comments:

The authors combined theoretical and experimental investigations to characterize pneumonia and its comorbidities and to assess the host immune response measured by TREC/KREC levels in patients with pneumonia. The experimental study found that community-acquired pneumonia patients had higher TREC levels, while nosocomial pneumonia (NP) patients had lower TREC/KREC levels compared to controls. Low TREC and KREC levels were predictive for the development of NP, ARDS, sepsis, and lethal outcome. The authors conclude that TREC/KREC analysis can be considered as a potential prognostic test in patients with pneumonia.

General concerns:

1.      Please describe the rationale to perform 1. theoretical and 2. experimental studies.

2.      The p values were very small in the Table 1 and the figures. Please confirm those p values are correct in Tables and Figures.

3.      The bronchoalveolar lavage (BAL) fluids were obtained from all CAP and NP patients and 94% and 65% of them were not mechanically ventilated. Is it mandatory to obtain BALF in these patients though these patients signed the consent forms?

4.      2. Materials and Methods: 2.1. Theoretical study with Columbia Open Health Data (COHD): Exclusion criteria for the CAP and NP groups included lack of informed consent, conditions related to the immune system (infections, anaphylactic conditions, autoimmune diseases), chronic respiratory diseases, malignancy, decompensated chronic conditions, extensive bleeding and pregnancy. However, one patient in NP group had benign neoplasms. Please clarify this?

5.      Please describe the association of TREC and KREC levels with respiratory pathogens.

Author Response

Dear reviewer, we are grateful to you for careful reading, important considerations and constructive comments. Please find enclosed our replies, comments and point-by-point description of the changes made to the manuscript in response.

Reviewer #1

We are grateful to the reviewer for careful reading, important considerations and constructive comments.

General concerns:

  1. Please describe the rationale to perform 1. theoretical and 2. experimental studies.

Reply

We included the rationale in the text of the last section of Introduction (lines 113-130):  «Pneumonia development, its severity and outcome depend on age, comorbidities and host immune response [1]. Our work is aimed at exploring the relationships between these conditions. We combined theoretical and experimental studies to characterize pneumonia with its comorbidities and age distributions and to evaluate the diagnostic utility of TRECs/KRECs as markers of immune response in patients with CAP and NP. Evidence for the association of comorbidities with pneumonia is based on studies generally focused on specific classes of diseases and specific age groups [1,13,14,16]. These data often vary greatly and do not provide a holistic view of the entire spectrum of pneumonia-related conditions. Optimal knowledge acquisition can be achieved using open access data aggregating information for large cohorts of patients. An example of such a resource is the Columbia Open Health Data (COHD) resource, which provides access to clinical concept prevalence and co-occurrence from electronic health records (EHR) [27]. The theoretical part of the work was performed using COHD.

  1. The p values were very small in the Table 1 and the figures. Please confirm those p values are correct in Tables and Figures.

Reply

We have carefully checked all statistical data and confirm their correctness.

We added one sentence to the captions to Table 1 (lines 230-231). “The significance of microbiological data was calculated for a subgroup of patients with identified pathogens”.

  1. The bronchoalveolar lavage (BAL) fluids were obtained from all CAP and NP patients and 94% and 65% of them were not mechanically ventilated. Is it mandatory to obtain BALF in these patients though these patients signed the consent forms?

Reply

Diagnosis and treatment were carried out in accordance with the current hospital protocols developed on the basis of national recommendations (ref. 32, 33). No invasive procedures have been carried out for scientific research.

  1. Materials and Methods: 2.1. Theoretical study with Columbia Open Health Data (COHD): Exclusion criteria for the CAP and NP groups included lack of informed consent, conditions related to the immune system (infections, anaphylactic conditions, autoimmune diseases), chronic respiratory diseases, malignancy, decompensated chronic conditions, extensive bleeding and pregnancy. However, one patient in NP group had benign neoplasms. Please clarify this?

Reply

We adhered to the definition of a benign tumor as unrelated to malignant neoplasms https://www.cancer.gov/publications/dictionaries/cancer-terms/def/neoplasm.

  1. Please describe the association of TREC and KREC levels with respiratory pathogens.

Reply

Thank you for this interesting suggestion. We included these data in Supplementary Figure S2.

Section 2.2.2. We added the sentence (lines 259-263) “Analysis of TRECs/KRECs in the groups of patients stratified by pathogens (CAP: Streptococcus pneumoniae, Staphylococcus aureus and Haemophilus influenzae; NP: Pseudomonas aeruginosa, Klebsiella pneumoniae and Acinetobacter baumannii) did not reveal differences in the distribution of TRECs/KRECs depending on the pathogen (Supplementary Figure S2)”.

Section 3. Discussion. We added the sentence (lines 331-333) “Our results of the experimental part of the study indicate the absence of pathogen-specific differences in the distribution of TRECs and KRECs. This observation seems reasonable since…”

Section 3. Study limitations. We added the sentence (lines 372-374) “Due to the high frequency of mixed infections, the comparison groups for assessing the pathogen-specific distribution of TRECs/KRECs included both patients with single and mixed infections”.

Reviewer 2 Report

I appreciate a lot the manuscript of Kashatnikova and colleagues. I find it well wrote, high quality presentation and with an important research question

below my few suggestions:

1. Introduction. very well done. I suggest to add the issue that during covid 19 pandemic pneumonia registered also an increase of antibiotic resistance and this is an important issue when treat pneumonia (see and cite Impact of SARS-CoV-2 Epidemic on Antimicrobial Resistance: A Literature Review. Viruses. 2021 Oct 20;13(11):2110. )

2. Methods and results: high level,congratulations

3. Discussion: underline the role of antibiotic resistance as global health problem and which strategy can be applied to reduce spread of AMR.

IN addition discuss the role of different pathogens and which remain the difficult to treat as showed in other interesting focus paper

Furthoremore, give some global health problem that came from your very interesting paper  

Author Response

Dear reviewer, we thank you for positive comment and useful recommendations. Please find enclosed our replies, comments and point-by-point description of the changes made to the manuscript in response.
